# Prevalence and Clinical Characteristics of Dyssynergic Defecation and Slow Transit Constipation in Patients with Chronic Constipation

**DOI:** 10.3390/jcm10092027

**Published:** 2021-05-09

**Authors:** Samuel Tanner, Ahson Chaudhry, Navneet Goraya, Rohan Badlani, Asad Jehangir, Dariush Shahsavari, Zubair Malik, Henry P. Parkman

**Affiliations:** Section of Gastroenterology, Department of Medicine, Lewis Katz School of Medicine at Temple University, Philadelphia, PA 19140, USA; Samuel.Tanner@uhhospitals.org (S.T.); achaudhry@temple.edu (A.C.); tug69854@temple.edu (N.G.); rohan.badlani@temple.edu (R.B.); asadjehangir@gmail.com (A.J.); Dariush.Shahsavari@tuhs.temple.edu (D.S.); zubair.malik@tuhs.temple.edu (Z.M.)

**Keywords:** dyssynergic defecation, constipation, slow transit constipation, motility, colonic transit, anorectal manometry, gastrointestinal disorders

## Abstract

Patients with chronic constipation who do not respond to initial treatments often need further evaluation for dyssynergic defecation (DD) and slow transit constipation (STC). The aims of this study are to characterize the prevalence of DD and STC in patients referred to a motility center with chronic constipation and correlate diagnoses of DD and STC to patient demographics, medical history, and symptoms. High-resolution ARM (HR-ARM), balloon expulsion testing (BET) and whole gut transit scintigraphy (WGTS) of consecutive patients with chronic constipation were reviewed. Patients completed questionnaires describing their medical history and symptoms at the time of testing. A total of 230 patients completed HR-ARM, BET, and WGTS. Fifty (22%) patients had DD, and 127 (55%) patients had STC. Thirty patients (13%) had both DD and STC. There were no symptoms that were suggestive of STC vs. DD; however, patients with STC and DD reported more severe constipation than patients with normal transit and anorectal function. Patients with chronic constipation often need evaluation for both DD and STC to better understand their pathophysiology of symptoms and help direct treatment.

## 1. Introduction

Constipation is a common disorder in Americans: primary constipation, e.g., constipation that is not secondary to another underlying disease or medication, is present in 12–17% of the population [1]. These patients may fail medical therapy and are referred to gastroenterology for evaluation and management. History and physical examination have been suggested to be poor predictors of underlying pathophysiology; further assessment of these patients is often suggested for two common etiologies–slow colon transit and dyssynergic defecation [2].

Two techniques commonly used in the evaluation of patients with constipation are anorectal manometry (ARM) to assess pelvic floor function and colonic transit testing [3]. In ARM, rectal pressures are recorded during balloon inflation and deflation as well as during a simulation of defecation with balloon distension in the rectum, which are used to describe defecation patterns in patients. This test is then used in conjunction with other testing such as balloon expulsion testing and/or defecography to identify underlying anorectal floor pathophysiology, such as dyssynergic defecation, and help direct treatment such as biofeedback therapy [4]. Assessment of colonic transit can be performed by radioopaque markers, wireless motility capsule, or scintigraphy. Whole gut transit scintigraphy (WGTS) assesses colonic transit as well as gastric emptying and small bowel transit [5]. The colonic transit result is often characterized as normal transit or a severe colonic transit abnormality suggesting colonic inertia, generalized colonic transit delay, and functional rectosigmoid obstruction [6].

In general, patients with chronic constipation can have distinct phenotypes, including slow transit constipation (as assessed by WGTS or other colonic transit study), dyssynergic defecation (as assessed by ARM and other tests of anorectal function), a combination of slow transit constipation and dyssynergic defection, no slow transit constipation or dyssynergic defecation (i.e., normal colonic transit and normal anorectal function). However, few studies have examined the prevalence of these phenotypes or the association of these diagnoses with findings of WGTS and HR-ARM. Moreover, the literature has conflicting data on the association between symptoms and underlying pathophysiology of constipation [7,8]. The aims of this study were to: (1) Assess the prevalence of DD and STC in patients referred to a motility center with chronic constipation; (2) Correlate diagnoses of DD and STC to patient demographics, medical history, and symptoms.

## 2. Materials and Methods

### 2.1. Patients

We retrospectively analyzed consecutive patients who underwent whole gut transit scintigraphy and high-resolution anorectal manometry for the evaluation of chronic constipation at Temple University Hospital Motility Center between 1 January 2016 and 31 December 2019. Exclusion criteria included history of surgery on the GI tract, pregnancy, age <18 years, inability to complete testing, or a primary indication other than constipation for testing (such as abdominal pain).

### 2.2. Questionnaires

Patients were asked to fill out several questionnaires at the time of testing. These questionnaires assessed the patient’s demographic profile, current medications, medical and surgical history, upper GI symptom severity via the Patient Assessment of Upper Gastrointestinal Disorders-Symptom Severity Index (PAGI-SYM) with four additional domains including constipation, diarrhea, belching, and flatulence [9]. Symptoms over the prior two weeks were graded by the patient at the time of colonic transit scintigraphy from none = 0 to very severe = 5. The 20-item PAGI-SYM questionnaire was used to calculate composite scores of six symptom domains. The Rome III diagnostic questionnaire for lower GI disorders was also used [10]. Additionally, the frequency of habits related to the patient’s bowel movements were also assessed.

### 2.3. Balloon Expulsion Testing

Balloon expulsion testing was performed [11]. A 4 cm balloon was inserted into the rectum and filled with 50 cc of water. The patient was then sent to the bathroom and asked to measure the amount of time with a stopwatch it took to expulse the balloon. Abnormal balloon expulsion was defined as longer than 60 s [12].

### 2.4. High-Resolution Anorectal Manometry

High-resolution anorectal manometry (HR-ARM) was performed [13,14,15]. A 4.2 mm diameter solid-state catheter consisting of 12 circumferential sensors (10 sensors at 6 mm intervals along the anal canal and 2 sensors in the rectal balloon was used to measure pressure profiles, reflexes, and sensation in the anorectal region (Medtronic, Inc., Shoreview, MN, USA). The patient was placed in the left lateral decubitus position with their knees flexed. The catheter was inserted and advanced until the high-pressure zone of the internal anal sphincter was localized. This was followed by a 2-minute period of stabilization to allow anal tone to return to baseline. Each patient was then asked to squeeze the anus 2 times for 20 s at a time to simulate holding in a stool (volitional contraction) to measure volitional external anal sphincter contraction pressure. Graded balloon distension testing was performed by measuring the basal anal sphincter pressure and then inflating the rectal balloon by 10 mL to first sensation point then intervals of 30 mL to each subsequent sensation point (desire to defecate, urge to defecate, and maximum tolerance). The presence of the rectoanal inhibitory reflex (relaxation of the internal anal sphincter during rectal distension [RAIR]) was also recorded. The patient was then asked to bear down 3 times (20 s each time) to simulate defecation. During bear down maneuvers, the intrarectal pressure and internal sphincter percent relaxation were recorded. Internal anal sphincter percent relaxation was defined as the ratio of amount of anal relaxation to anal resting pressure × 100 [11]. Dyssynergic defecation was defined as an abnormal balloon expulsion test as well as an abnormal pattern of defecation identified on anorectal manometry by a combination of either incomplete relaxation or paradoxical contraction of the anal sphincter with either inadequate or adequate generation of intra-rectal pressure during bear-down maneuver [16].

### 2.5. Whole Gut Transit Scintigraphy

Whole gut transit scintigraphy was performed [6,17]. Patients stop any constipation medications for 3 days prior to the study and come in fasting after midnight. Patients consumed a dual-isotope test meal consisting of an egg beater meal labeled with 500 uCi of Tc-99m sulfur colloid added to the egg white portion of the meal. The meal also consisted two pieces of white-bread toast and jam. The liquid portion of the meal consisted of 100 uCi of In-111 DTPA in 6 oz of water. Imaging occurred at 0, 0.5, 1, 2, 3, 4 h to evaluate for gastric emptying, and at 5 and 6 h to evaluate for small bowel transit. Images were then obtained at 24, 48, and 72 h after meal ingestion to determine colonic transit by evaluating geometric centers of colonic activity [18].

Gastric emptying was quantified as the percentage of meal remaining in the stomach region of interest, with delayed gastric emptying defined as >10% of meal remaining at 4 h. Small bowel transit was defined as the percentage of meal remaining prior to the ileocecum at 6 h, with >40% defined as normal small bowel transit. For the colonic images, counts were measured in regions of interest corresponding to the cecum/ascending colon (region 1), hepatic flexure (region 2), transverse colon (region 3), splenic flexure (region 4), descending colon (region 5), and rectosigmoid (region 6). Administered radioactivity that was unaccounted for in the images was assumed to have been eliminated by bowel movements and was designated as region 7. The geometric center for colonic activity was calculated as the summation of scintigraphic counts at each region of interest as a fraction of the total counts, weighted by that region’s assigned number. Slow transit constipation was defined as a geometric center ≤4.6 at 48 h [19].

### 2.6. Data Analysis

For multi-population comparisons of continuous variables, the ANOVA test followed by pairwise t tests were used for normally distributed data while the Kruskal–Wallis test followed by Dunn method was used for non-normally distributed data. The Bonferroni correction was applied to adjust for multiple comparisons. Categorical variables were compared using Fisher’s exact test. For multi-population testing using Fisher’s exact test, post hoc adjusted residuals were calculated. Statistical testing was executed using Microsoft Excel (Microsoft Corp, Redmond, WA, USA) and SPSS (IBM Corp, Armonk, NY, USA).

## 3. Results

A total of 230 patients completed WGTS, BET and HR-ARM and met inclusion criteria for this study. The mean age of our cohort was 47.5 years, comprised of 89% women, and had a mean BMI of 25.9 kg/m^2^. The median duration of constipation symptoms was 2.0 years (interquartile range, 1.0 to 5.0 years).

### 3.1. Pathophysiology Using BET, HR-ARM and Colonic Transit Scintigraphy

Of the 230 patients, 20 patients (9%) had dyssynergic defecation, 97 patients (42%) had slow transit constipation, 30 patients (13%) had both dyssynergic defecation and slow transit constipation, and 83 patients (36%) had neither dyssynergic defecation nor slow transit constipation (Figure 1 and Table 1). In total, 50 patients had dyssynergic defecation, of whom 30 (60%) also had slow transit constipation. Conversely, 127 patients had slow transit constipation, of whom 30 (24%) also had dyssynergic defecation.

### 3.2. Findings on Whole Gut Scintigraphy and Anorectal Manometry

There were differences in gastric emptying between the populations at both 2 and 4 h (*p* = 0.04 and *p* < 0.01, respectively) (Table 1). Follow-up testing showed that these differences of gastric emptying at 2 h existed between populations that had a diagnosis of STC vs. no STC (e.g., STC only vs. DD only, STC and DD vs. DD only, STC only vs. no STC or DD, STC and DD vs. no STC or DD, all *p* < 0.001). Similarly, gastric emptying at 4 h was different based on the presence of STC (all *p* < 0.001). No differences were seen in small bowel transit. Colonic transit at 24, 48, and 72 h all differed among the populations (*p* < 0.001).

On HR-ARM, mean resting pressure differed among the populations (*p* = 0.03). Patients with STC only had lower mean resting pressure compared to patients with no STC or DD (65.2 ± 2.1 vs. 72.1 ± 2.3, *p* = 0.03) and there was a trend towards significance in patients with STC only vs. DD only (65.2 ± 2.1 vs. 78.7 ± 6.5, *p* = 0.06). There were also statistically significant differences in anal relaxation on simulated defecation (*p* < 0.001). These differences were seen in populations with DD vs. no DD (e.g., STC only vs. DD only, STC + DD vs. STC only, DD vs. no STC or DD, STC + DD vs. no STC or DD, all *p* < 0.001).

On rectal sensory testing, there were differences on first desire to defecate, first urge to defecate, and maximum tolerance (*p* < 0.01, *p* < 0.01, and *p* = 0.04, respectively). There were differences in first desire to defecate in patients with STC only vs. no STC or DD (64.6 ± 4.0 vs. 49.9 ± 3.9, *p* < 0.001) and STC and DD vs. no STC or DD (85.5 ± 11.1 vs. 49.9 ± 3.9, *p* < 0.01). There are statistically significant differences in first urge to defecate in STC + DD vs. STC only (135.0 ± 10.4 vs. 109.0 ± 4.7, *p* = 0.04), STC + DD vs. no STC or DD (135.0 ± 10.4 vs. 93.5 ± 4.5, *p* < 0.001), and STC only vs. no STC or DD (109.0 ± 4.7 vs. 93.5 ± 4.5, *p* = 0.02). In maximum tolerance, patients with STC + DD had higher thresholds than patients with no STC or DD (162.1 ± 9.6 vs. 134.4 ± 4.9, *p* = 0.01).

### 3.3. Demographics, Medical History, and Symptoms

There was a difference in genders among the different populations (*p* = 0.02). Patients with STC (with or without DD) were more likely to be female than patients without STC (95% vs. 83%, *p* < 0.01). Otherwise, there was no statistically significant difference in age, BMI, race, or medical history.

Symptomatically, there were differences in the average number of bowel movements per week among the different populations (*p* < 0.001). Patients with either STC + DD or STC only had significantly fewer BMs than patients without STC or DD (both *p* < 0.001). There were also differences in self-reported severity of constipation (*p* < 0.02). Statistically significant differences exist between patients with STC only or STC + DD and patients with no STC or DD (4.4 ± 0.1 vs. 3.9 ± 0.2, *p* = 0.03 and 4.7 ± 0.1 vs. 3.9 ± 0.2, *p* < 0.01, respectively). There were also differences in the severity of intermittent diarrhea (*p* < 0.01), including statistically significant differences between STC + DD and DD only (0.8 ± 0.3 vs. 1.8 ± 0.5, *p* = 0.05), STC + DD and no STC or DD (0.8 ± 0.3 vs. 1.4 ± 0.2, *p* = 0.05), and STC only and no STC or DD (0.6 ± 0.2 vs. 1.4 ± 0.2, *p* < 0.01). There were no statistically significant differences in upper GI symptoms or differences in bowel habits.

## 4. Discussion

This study describes the prevalence of dyssynergic defecation and slow transit constipation in patients referred to an academic medical center with chronic constipation. Both of these pathophysiological causes were common, with dyssynergic defecation present in 22% of patients and slow transit constipation present in 55% of patients. In this study, only 36% of patients with constipation had normal test results for colonic transit and anorectal coordination (no abnormality in either BET or HR-ARM); most patients (64%) had defined abnormalities of transit and/or defecation explaining their symptoms. Importantly, 13% of all patients had evidence of both dyssynergic defecation and slow transit constipation. This suggests that there may be more than one underlying cause for their constipation—both dyssynergic defecation and slow colonic transit.

Previous studies have had conflicting data regarding the prevalence of slow transit constipation and dyssynergic defecation. We previously reported that 67% of patients presenting with chronic constipation to our center had a colonic transit disorder and 37% had dyssynergic defecation [20]. This contrasts with a study of 1009 patients who underwent both pelvic floor function testing and scintigraphy, where only 7% of patients were found to have slow transit constipation and 27% had pelvic floor dysfunction [21]. A possible explanation for the variance in dyssynergic defecation prevalence in the literature is the differing criteria and diagnostic testing used to define dyssynergic defecation. For example, in our previous study, DD was defined as abnormalities in 2 of 4 of the following tests: ARM, electromyography, BET, and defecography. In contrast, the latter study defined dyssynergic defecation as abnormal BET plus high anal sphincter pressure and/or failure of anorectal angle to open ≥15° between resting and straining. A challenge with applying multiple diagnostic tests to define DD is that there can often be poor agreement between them [22]. This study used the newly proposed London Classification, a consensus agreement among the international anorectal physiology working group (IAPWG), which defines dyssynergic defecation as abnormal BET and anorectal coordination on ARM (although they do concede that additional testing may be needed if a patient has either abnormal BET with normal ARM or normal BET with abnormal ARM if there is clinical suspicion for DD) [16]. However, our study uses the strict definition of abnormal BET and ARM, which may account for the lower prevalence of DD in our population than our previous study (22% vs. 37%) and may underestimate the true number of patients with DD.

Regardless, our study illustrates that some patients can have both dyssynergic defecation and delayed colonic transit. Previous studies have shown that there is an overlap between dyssynergic defecation and slow transit constipation [13,23,24,25]. Why some patients have both disorders is not clear. One study demonstrated that slow transit constipation improved after the completion of biofeedback therapy, suggesting that an abnormal colonic transit test may be the result of dyssynergic defecation rather than suggestive of colonic inertia or generalized slow transit constipation [13]. There may be two explanations for this overlap of both findings in the same patient. First, the study used a protocol where colonic transit was measured by the number of retained radioopaque markers at 120 h after ingestion. This does not account for the location of these markers, and many may be accumulated in the rectosigmoid region secondary to poor defecatory mechanics despite normal colonic transit. However, a large multicenter study suggested that the location of markers in the rectosigmoid region is not correlated with dyssynergic defecation [26]. A second explanation is that dyssynergic defecation and slow transit constipation may be linked. A study by *Nullens* et al., showed that dyssynergic defecation is associated with delayed overall colonic transit at 48 h [27]. It has further been suggested that dyssynergic defecation can lead to a reflex inhibition of colonic transit in the proximal colon [28,29]. While this is certainly feasible, this study showed no difference in gastric, small bowel, or colonic transit between patients with dyssynergic defecation and patients without dyssynergic defecation, regardless of whether the patient met diagnostic criteria for slow transit constipation.

This study also examined symptoms and patients’ demographics associated with dyssynergic defecation and slow transit constipation. Interestingly, there was no differences in bowel habits between the different populations (STC + DD, STC only, DD only, no STC or DD). This contrasts with previous studies which suggested patients with dyssynergic defecation may be associated with patients using digital maneuvers to complete a bowel movement, excessive straining, a feeling of incomplete evacuation, the passage of hard stools, and infrequent stooling [9,30]. While these symptoms were common in our patient population, they were not unique to patients who had objective evidence on dyssynergic defecation by BET and HR-ARM testing. Conversely, no symptoms were suggestive of slow transit constipation. However, there were differences in the severity of constipation and diarrhea experienced by patients. Patients with STC + DD or STC only had more significant constipation than patients with no abnormalities as assessed by bowel movements per week. The data are also suggestive that patients with DD only had more severe constipation than patients with no abnormalities, although this did not reach statistical significance, perhaps due to small sample size of this population. Self-reported severity of diarrhea was also lower in the STC + DD population compared to DD only or no abnormalities group. However, we do not believe this is clinically significant as most patients only reported none or mild diarrhea.

We also show that slow transit constipation is associated with delayed gastric emptying at 2 and 4 h. This contrasts with a previous study by our group which suggested that both slow transit constipation and dyssynergic defecation was associated with delayed gastric emptying [20]. This study builds on previous studies by assessing upper gastrointestinal symptoms to determine whether upper GI symptoms are prevalent in patients with slow transit constipation. There were no differences in upper GI symptoms, which suggests that findings of delayed gastric emptying in slow transit constipation has unclear clinical significance. Further studies are needed to determine clinical significance of delayed gastric emptying in patients presenting with chronic constipation. Further studies are also needed to investigate the correlation of rectal sensitivity testing to STC and DD as both STC and DD were associated with higher thresholds for rectal distension. This would suggest rectal hyposensitivity, which might also be playing a role in their symptoms of constipation.

An important implication of this study is the diagnostic approach to patients who present with chronic constipation. Currently, both the American Gastroenterology Association (AGA) and American College of Gastroenterology (ACG) guidelines are to start testing with anorectal manometry [31,32]. If ARM reveals a defecatory disorder, then treatment of this, such as biofeedback therapy, should be pursued [32]. However, it is unclear whether biofeedback therapy would also correct slow transit constipation. There have been few studies that have assessed biofeedback therapy in slow transit constipation, with varying reports in effectiveness [28,33,34]. However, all these studies suffer from low or very low quality of evidence as noted in a Cochrane review on biofeedback therapy in chronic constipation [35]. Symptom profiles among patients with STC + DD compared to singular diagnoses were largely similar. Given that symptom profiles cannot distinguish STC, DD, and STC + DD, it is important that patients undergo testing for both constipations to guide treatment options. The primary treatment modality for dyssynergic defecation is biofeedback therapy [36,37]. In contrast, slow transit constipation that has failed laxative therapy has limited treatment options, and severe cases may require surgery, such as total colectomy [38,39]. The impact of concomitant STC in patients undergoing treatment for DD has not been well-studied. A study of 52 patients found biofeedback therapy more effective in patients with pelvic floor dyssynergia and slow transit constipation compared to patients with slow transit constipation only [28]. However, that study did not compare pelvic floor dyssynergia with slow transit constipation to pelvic floor dyssynergia as all patients included in the study met criteria for slow transit constipation [28]. We did not look at treatment outcomes for treating dyssynergic defecation and/or slow transit constipation. This would be of particular interest in the patients with both disorders to better understand what type of treatment works best, and if the delayed colonic transit normalizes in patients with slow transit constipation and DD with treatment of the DD.

Our study evaluates a large number of patients with chronic constipation, undergoing the state-of-the-art tests–HR-ARM (with BET) and whole gut transit scintigraphy. Validated questionnaires were used to assess their symptoms. The Rome III criteria were used to help characterize the constipation symptoms. However, there are several limitations to this study. First, most patients were referred to our tertiary academic center and, therefore, do not represent the broader community with constipation. Given that patients had failed medical therapy for constipation, there may be a higher proportion of functional disorders such as dyssynergic defecation and slow transit constipation. Second, our symptom survey was limited to symptoms patients had been experiencing over the past several weeks. Other studies have looked at associations between functional constipation and childhood traumas, such as physical or sexual abuse [40]. Another limit of only looking at symptoms in the past several weeks is that many patients were on some sort of laxative therapy, which may have affected constipation symptoms.

## 5. Conclusions

This study describes the prevalence of dyssynergic defecation and slow transit constipation in patients referred with chronic constipation. In our study, only 36% of patients had normal test results for colonic transit and HR-ARM; most patients (64%) had defined abnormalities of transit and/or defecation explaining their symptoms. Dyssynergic defecation found in 22% of patients and slow transit constipation found in 55% of patients. Importantly, 13% of all patients had evidence of both dyssynergic defecation and slow transit constipation. Symptoms alone were a poor predictor of underlying pathophysiology of constipation. Thus, to get a proper evaluation of the pathophysiology of a patient’s constipation from a motility standpoint, both ARM and colonic transit need to be assessed, as both are common, including both disorders in the same patient.

## Figures and Tables

**Figure 1 jcm-10-02027-f001:**
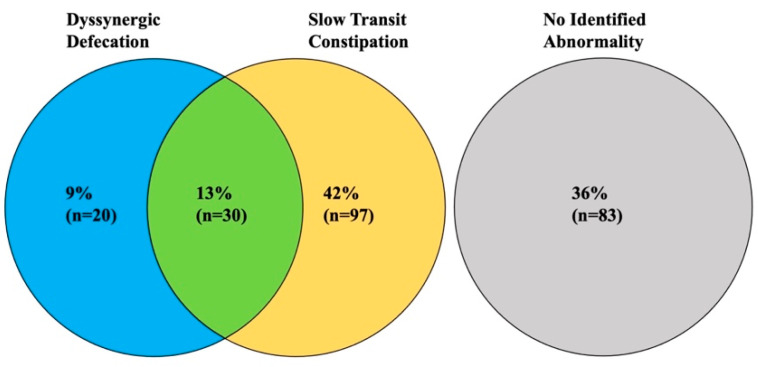
Prevalence of DD and STC in patients presenting to our center for chronic constipation.

**Table 1 jcm-10-02027-t001:** Comparison of patients with combined vs. singular diagnoses (slow transit constipation and dyssynergic defecation vs. slow transit constipation and normal anorectal function or dyssynergic defecation and normal colonic transit or normal anorectal function and normal colonic transit).

	STC + DD	STC Only	DD Only	No STC or DD	*p*-Value
***n***	**30**	**97**	**20**	**83**	**--**
**Demographics**					
Age (mean ± SE)	49.9 ± 3.5	48.0 ± 1.5	47.4 ± 2.3	46.4 ± 1.9	0.88
Gender (% female)	97%	94%	85%	81%	0.02
BMI (mean ± SE)	26.3 ± 1.5	24.9 ± 0.7	25.3 ± 1.0	27.1 ± 0.7	0.10
Race White Black Other Unknown	87% 0% 10% 3%	84% 4% 7% 5%	80% 15% 5% 0%	81 %1% 12% 6%	0.16
**Past Medical History**					
Diabetes	7%	13%	30%	16%	0.17
Anxiety or Depression	20%	23%	45%	29%	0.18
Other psych	10%	12%	10%	7%	0.72
GERD	13%	23%	20%	20%	0.56
Thyroid Disease	13%	14%	10%	6%	0.29
Connective tissue disease	0%	5%	10%	6%	0.39
**Whole Gut Transit Scintigraphy**					
Gastric Emptying					
2h (%)	41.9 ± 3.2	47.4 ± 2.0	43.2 ± 4.5	38.2 ± 2.2	0.04
4h (%)	13.0 ± 1.9	15.9 ± 1.8	12.5 ± 3.3	9.6 ± 1.3	<0.01
% Delayed	48%	45%	35%	29%	0.10
Small Bowel Transit (% Delayed)	20%	27%	35%	22%	0.55
Colonic Transit (GC)					
24h	3.1 ± 0.1	3.1 ± 0.1	4.8 ± 0.3	4.9 ± 0.1	<0.001
48h	3.6 ± 0.1	3.6 ± 0.1	6.0 ± 0.2	6.1 ± 0.1	<0.001
72h	4.2 ± 0.2	4.0 ± 0.1	6.4 ± 0.1	6.5 ± 0.1	<0.001
**Anorectal Manometry**					
Mean resting pressure (mmHg)	70.7 ± 3.6	65.2 ± 2.1	78.7 ± 6.5	72.1 ± 2.3	0.03
Maximal squeeze pressure (mmHg)	129.1 ± 11.1	130.5 ± 6.6	135.5 ± 11.4	136.7 ± 7.1	0.82
Intrarectal pressure (mmHg)	51.0 ± 4.1	53.3 ± 2.6	63.5 ± 7.5	61.1 ± 3.7	0.27
% Anal Relaxation	7.6 ± 3.5	23.1 ± 2.3	4.6 ± 5.3	19.3 ± 3.1	<0.001
**Rectal Sensitivity Testing**					
First sensation (mL)	41.3 ± 8.5	24.0 ± 2.2	25.0 ± 5.5	21.0 ± 1.9	0.19
First desire (mL)	85.5 ± 11.1	64.6 ± 4.0	58.7 ± 8.2	49.9 ± 3.9	<0.01
First urge (mL)	135.0 ± 10.4	109.0 ± 4.7	117.4 ± 12.1	93.5 ± 4.5	<0.01
Maximum tolerance (mL)	162.1 ± 9.6	145.3 ± 4.9	158.9 ± 12.0	134.4 ± 4.9	0.04
RAIR (% not present)	0%	5%	10%	6%	0.44
Abnormal BET	100%	94%	100%	93%	0.99
**Symptoms**					
Duration (median, IQR [years])	1 (1–3)	3 (1–11)	2 (1–10)	2 (1–3)	0.08
BMs per week	1.3 ± 0.2	2.0 ± 0.2	2.4 ± 1.0	4.9 ± 0.6	<0.001
Abdominal Pain (1d/wk or greater)	100%	96%	100%	97%	0.99
Urinary leakage (1d/wk or greater)	27%	33%	18%	17%	0.23
Fecal leakage (1d/wk or greater)	5%	5%	0%	8%	0.90
Fecal urgency (1d/wk or greater)	14%	18%	45%	24%	0.18
**Symptom Severity ^1^**					
Constipation	4.7 ± 0.1	4.4 ± 0.1	4.5 ± 0.2	3.9 ± 0.2	0.02
Diarrhea	0.8 ± 0.3	0.6 ± 0.2	1.8 ± 0.5	1.4 ± 0.2	<0.01
Belching	2.2 ± 0.4	2.2 ± 0.2	3.1 ± 0.5	2.6 ± 0.2	0.27
Flatulence	2.3 ± 0.3	2.6 ± 0.2	2.8 ± 0.4	2.5 ± 0.2	0.68
Regurgitation and heartburn	1.5 ± 0.2	1.5 ± 0.2	2.2 ± 0.4	1.6 ± 0.2	0.42
Fullness and early satiety	3.6 ± 0.2	3.1 ± 0.2	3.4 ± 0.3	3.0 ± 0.2	0.26
Nausea & vomiting	2.0 ± 0.3	1.7 ± 0.2	2.6 ± 0.5	1.6 ± 0.2	0.20
Bloating	3.6 ± 0.3	3.4 ± 0.2	4.2 ± 0.3	3.3 ± 0.2	0.21
Upper abdominal pain	3.1 ± 0.2	2.8 ± 0.2	3.6 ± 0.2	2.8 ± 0.2	0.45
Lower abdominal pain	2.7 ± 0.3	2.8 ± 0.2	3.5 ± 0.3	2.8 ± 0.2	0.42
**Bowel Habits ^2^**					
Hard or lumpy stools	65%	68%	73%	55%	0.43
Straining	78%	81%	92%	79%	0.83
Feeling of incomplete evacuation	83%	81%	92%	79%	0.87
Sensation that stool could not be passed (blocked)	68%	71%	80%	66%	0.81
Press on or around bottom or remove stool to complete BM	36%	31%	27%	27%	0.84
Difficulty “letting go” to allow stool to come out during BM	59%	44%	27%	45%	0.38

^1.^ Based on a 5-point scale from 0 (none or absent) to 5 (very severe). Results expressed as mean ± standard error. ^2.^ Percentage of patients responding often, most of the time, or always on scale of never, rarely, sometimes, often, most of the time, always.

## Data Availability

Data supporting reported results are available through reasonable request to the authors.

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
