# Peer review of "Prevalence and Clinical Characteristics of Dyssynergic Defecation and Slow Transit Constipation in Patients with Chronic Constipation"

_jcm, 2021, doi:10.3390/jcm10092027_

Round 1
Reviewer 1 Report
Thanks for addressing my comments.
Author Response
Thank you for reviewing our manuscript.
Reviewer 2 Report
The revised article is very much improved. However, there are a few points i would like the authors to respond to:
- The numbers have changed dramatically. Why is that? Please explain to the reviewers.
- The discussion does not address reference 28 which tests the results of biofeedback training in patients with slow transit constipation vs. those with slow transit plus dyssynergia. This was a treatment study which showed greater improvements in the dyssynergic defecation subgroup based on both manometric findings and self-reported improvement.
- The authors argue that it is unknown whether slow transit constipation is improved by biofeedback. This is controversial as there are articles on both sides of the issue, but the 2006 article by Chiarioni should be mentioned as it claims that biofeedback helps those with dyssynergia but not those with slow transit. This is an area of interpreting the findings, but the fact that the answer is likely to challenge the opinions of two major GI organizations makes it critical.
- The authors argue that symptoms are not helpful in distinguishing subtypes of constipation. However, this is disputed by some of their own work. Symptoms are less reliable than manometric findings but are nevertheless important, especially when the outcome is a recommendation is for a problematic surgery versus a biofeedback procedure which is not harmful.
Author Response
Please see the attachment.

This manuscript is a resubmission of an earlier submission. The following is a list of the peer review reports and author responses from that submission.
Round 1
Reviewer 1 Report
The shift from Rome III to IV has redefined the concept of functional defecation disorders.
The definition of 'dyssynergic defecation' given by the authors, which is solely based on abnormal HRAM patterns of defecation is superseded by the need for at least 2 abnormal tests of evacuation according to the Rome IV criteria (i.e. BET, manometry, defecography).
In a seminal work, Grossi et al. (Gut 2018) demonstrated the poor diagnostic accuracy of manometry to discriminate between health and constipation.
Taken together, these considerations deeply undermine the validity of the results presented in this study.
Reviewer 2 Report
This is an ambitious observational study taken from 4 years of the authors practice. It uses state of the art techniques to measure gastric, intestinal, and anal manometry, and it uses the recently published London classification to define dyssynergic defecation. The aims of the study are (a) to show the prevalence of STC and DD in this population and (b) to correlate these diagnoses with medical history, symptoms, and physiology.
- The prevalence of STC and DD are reported, and they are similar to previous estimates. There are always minor differences depending on the referral pattern to the clinic. Most importantly the authors show that STC, DD, and an overlap of STC and DD are common.
- The demographic, medical history and physiology differ between groups, with worse symptoms of constipation in the STC+ DD group compared to the other groups. There are differences in age, sex, emptying rates in the stomach and colon, and symptoms of difficulty letting go of stools. However, the differences are not explained. Notice that the diagnoses overlap between groups.
- In the paper the authors say that showing treatment differences between groups would be important to influence practice. However, they do not report any treatment data. Is there data but that is not reported? Other studies have suggested that DD responds to biofeedback better than to laxatives, and that STC responds poorly to most treatments available to us.
- Tables 1, 2, and 3 overlap and it is not clear to me why they are all presented. I believe Table 3 contains the most granular detail of differences between groups.
- The relevance of these differences to treatment needs to be discussed in the discussion. Currently the differences are not mentioned.